# Instrument uncertainties of network-suitable groundbased microwave radiometers: overview, quantification, and mitigation strategies

Tobias Böck<sup>1</sup>, Moritz Löffler<sup>1,2</sup>, Tobias Marke<sup>1</sup>, Bernhard Pospichal<sup>1</sup>, Christine Knist<sup>3</sup>, and Ulrich Löhnert<sup>1,4</sup>

<sup>1</sup>Institute for Geophysics and Meteorology, University of Cologne, 50923 Cologne, Germany <sup>2</sup>German Meteorological Service (DWD), Technical Infrastructure and Operations, 14473 Potsdam, Germany

<sup>3</sup>German Meteorological Service (DWD), Meteorological Observatory Lindenberg – Richard Aßmann Observatory, 15848 Tauche OT Lindenberg, Germany

<sup>4</sup>Institute of Climate and Energy Systems, ICE-3: Troposphere, Forschungszentrum Jülich GmbH, 52428 Jülich, Germany

Correspondence: Tobias Böck (tobias-boeck@mail.de), Bernhard Pospichal (bernhard.pospichal@uni-koeln.de)

Abstract. To enhance observations of the Atmospheric Boundary Layer (ABL), the European Meteorological Network, EUMETNET, and the Aerosol, Clouds, and Trace Gases Research Infrastructure, ACTRIS, are currently collaborating to establish networks of MicroWave Radiometers (MWRs). MWRs can be used to derive thermodynamic profiles within the ABL. Understanding and assessing instrument uncertainties of state-of-the-art MWRs is therefore crucial for accurate observations and also data assimilation purposes. Some national weather services are currently exploring the potential of MWR networks to improve shortterm weather forecasts. In this paper, we discuss uncertainties inherent to the MWR instrument itself, namely (1) radiometric noise, (2) long-term drifts and jumps, (3) calibration repeatability, (4) biases/systematic differences between instruments, and (5) radome degradation. These uncertainties are expressed in brightness temperatures. For state-of-the-art MWRs (here, Generation 5 Humidity and Temperature PROfiler HATPRO-Gen5), radiometric noise at ambient temperatures is a maximum of 0.32 K in the V-band but usually lower, especially near cold load temperature ranges in the K-band ( $\leq 0.11$  K). Long-term drifts or jumps between calibrations, which are at least two months apart, are usually below 0.4 K in the K-band and 1.0 K in the V-band but can also be higher. Drifts do not follow a discernable timely pattern and are therefore not easily quantifiable in a meaningful way. Calibrating at least every six months is thus advised. Calibration repeatability is shown to be well under 0.16 K. Mean brightness temperature differences between two HATPRO-Gen5 instruments are shown to be as high as 0.15 K in the K-band and 0.58 K in the Vband at zenith viewing direction. The radome has significantly degraded due to weathering and needs to be replaced if, 10 min after a rain event, residual water on its surface still causes a temperature deviation of more than 2 K compared to a dry radome.

#### 1. Introduction and motivation








The atmospheric boundary layer (ABL) stands as a pivotal yet often under-sampled component of the atmosphere. Monitoring the ABL is highly important for short-range forecasting of severe weather events. Key atmospheric variables vital for Numerical Weather Prediction (NWP) applications, such as temperature (*T*) and humidity (*H*) profiles, currently present measurement challenges (Teixeira et al., 2021). Ground-

based microwave radiometers (MWRs) emerge as a promising solution for capturing *T* profiles within the ABL, complemented by providing coarse-resolution *H* profiles. HATPROs (Humidity And Temperature PROfilers) are by far the most widely used MWRs in Europe<sup>1</sup> and, therefore, warrant the need to derive reliable uncertainty budgets. These instruments offer continuous, unattended data collection across a broad spectrum of weather conditions (Liljegren, 2002; Rose et al., 2005; Cimini et al., 2011; Löhnert & Maier, 2012).

In addition to zenith observations that provide path-integrated parameters like integrated water vapor (*IWV*) and liquid water path (*LWP*) at high temporal resolutions (up to 1 s), elevation scans contribute to obtaining more precise temperature profiles within the ABL (Crewell & Löhnert, 2007). Furthermore, azimuth scans allow for evaluating horizontal variations in water vapor and cloud distribution (Marke et al., 2020). Previous studies, underscore the benefits of assimilating MWR observations into NWP models (De Angelis et al., 2016; Caumont et al., 2016; Martinet et al., 2020). However, this practice is currently routinely implemented only in the operational NWP model of MeteoSwiss (Vural et al., 2024).

Establishing an operational network of state-of-the-art MWRs is highly beneficial for enhancing meteorological observations, and several initiatives are currently dedicated to this objective. The European COST Action CA 18235 PROBE<sup>2</sup> (PROfiling the atmospheric Boundary layer at a European scale) (Cimini et al., 2020) and the European Research Infrastructure for the observation of Aerosol, Clouds, and Trace gases ACTRIS (Laj et al., 2024) have been working toward the establishment of continent-wide standards for MWR networks, serving both research and NWP applications. This also holds true for the GCOS (Global Climate Observing System)<sup>3</sup> Reference Upper-Air Network (GRUAN), which aims to provide climate data records not only at the surface of the Earth but throughout the atmospheric column. For that, GRUAN is considering implementing MWRs as well. Moreover, the E-PROFILE<sup>4</sup> program of EUMETNET<sup>5</sup> endorsed a business case to continuously supply MWR data to European meteorological services (Rüfenacht et al., 2021) for nowcasting purposes as well as assimilating MWR data in short-range models. The German Meteorological Service (DWD) is also exploring the potential of MWRs to enhance short-term weather forecasts across Germany.

For these endeavors to succeed, a comprehensive assessment of uncertainties associated with state-of-the-art MWRs is imperative. At the moment, a thorough analysis and overview of uncertainties related to these state-of-the-art instruments are hard to come by, especially for errors inherent to the MWR instrument like radiometric noise, drifts, calibration repeatability, biases, and radome degradation, which this study is scrutinizing in detail. While some aspects of these instrument uncertainties, as well as calibration uncertainties, have been discussed in earlier studies (Liljegren, 2002; Crewell & Löhnert, 2003; Maschwitz et al., 2013; Küchler et al., 2016), they warrant further investigation and updating to modern standards.

In addition to Böck et al. (2024), who analyzed measurement uncertainties from *external* sources like physical obstacles, horizontal inhomogeneities, instrument tilt, and radio frequency interference, this study tries to give a comprehensive overview of *internal* uncertainties of MWR. The aim is to develop a consistent error characterization that can be applied to any MWR network instrument. This will help the operators








<sup>&</sup>lt;sup>1</sup> See Cloudnet: https://instrumentdb.out.ocp.fmi.fi/ (last access: 20 October 2024)

<sup>&</sup>lt;sup>2</sup> https://www.cost.eu/actions/CA18235/ (last access: 28 November 2024)

<sup>&</sup>lt;sup>3</sup> https://www.gruan.org/network/about-gruan (last access: 20 October 2024)

<sup>&</sup>lt;sup>4</sup> EUMETNET Profiling Program

<sup>&</sup>lt;sup>5</sup> European Meteorological Network - https://www.eumetnet.eu/ (last access: 20 October 2024)

determine instrument malfunctions, perform system checks, and monitor instrument stability to decide when and if an intervention is necessary, such as an absolute calibration or a radome exchange. When analyzing internal MWR instrument uncertainties, it is important to remember that an absolute reference or the "truth" to which these errors can be compared is undisclosed and that all references are relative.

This study will first introduce the state-of-the-art HATPRO MWR instrument in Section 2. Section 3 will then describe the data sources and methods employed, forming the basis for analyzing instrument errors. This methodological segment will provide brief insights into calibration procedures, observation minus background (OmB) statistics, and a spectral consistency (SPC) retrieval, elucidating how these methodologies are applied. The main part of this study, encompassing the internal instrument errors of HATPRO MWRs, will be presented in Section 4. This section will specifically address radiometric noise, long-term instrument drifts or jumps between absolute calibrations, calibration repeatability, instances of systematic measurement differences between two instruments (referred to as biases here), and the degradation of radomes which determines how long a radome is still wet after a rain event (refer to Figure 1 for an overview of MWR instrument uncertainties). Section 5 provides recommendations and proposes mitigation strategies for MWR operators to minimize instrument uncertainties, while Section 6 summarizes all identified uncertainties and provides an outlook.

Figure 1: Sketch showing an overview of the analyzed MWR instrument uncertainties in terms of brightness temperatures (T<sub>B</sub>) over time. It depicts radiometric noise, long-term drifts or jumps between absolute calibrations, calibration repeatability, biases, and radome degradation. Biases usually disclose instrument differences to an absolute reference or theoretical truth but can also describe the differences between two instruments. Radome degradation is a measure of how long a wet radome needs to dry after a rain event.

# 2. Microwave radiometer instrument (HATPRO-Gen5)






In this study, we utilized state-of-the-art HATPROs of the fifth generation (Gen5). HATPROs are a prevalent type of MWR widely employed in Europe and manufactured by Radiometer Physics GmbH (RPG). These passive ground-based MWRs operate within the K-band and V-band spectra, covering the 22-32 GHz water vapor absorption line and the 51-58 GHz oxygen absorption complex. Operating with a temporal resolution on the order of seconds, HATPROs measure microwave radiances represented as T<sub>B</sub>S across 14 different frequency channels (refer to Table 1) simultaneously in zenith and various elevation angles. The retrieved  $T_{\rm BS}$  facilitate the determination of T profiles and H profiles, as well as path-integrated parameters such as IWV and LWP. The associated uncertainties for IWV and LWP retrievals have been shown to be below 0.5 kg m<sup>-2</sup> and 20 g m<sup>-2</sup>, respectively (Löhnert & Crewell, 2003). The accuracy of these retrievals depends upon the effectiveness of training and implementation processes. Elevation scans can be performed to increase the precision of T-profile retrievals within the ABL. Consequently, HATPROs can conduct measurements at multiple elevation angles, typically employing six to ten different angles ranging from 0° to 90°. While elevation scans are executed across all frequency channels, they are primarily utilized within the Vband channels due to their optically thick nature, especially in channels 10-14. This characteristic ensures a low penetration depth, enhancing the precision of temperature profile resolution when using varying elevation angles.

Table 1: Center frequencies and bandwidths of the 14 HATPRO channels (RPG-Radiometer Physics GmbH, 2022a).

|          | Channel | Center Frequencies | Bandwidth |  |
|----------|---------|--------------------|-----------|--|
|          | (ch) #  | [GHz]              | [MHz]     |  |
| K-band   | 1       | 22.24              | 230       |  |
|          | 2       | 23.04              | 230       |  |
|          | 3       | 23.84              | 230       |  |
| Curatan  | 4       | 25.44              | 230       |  |
| (water   | 5       | 26.24              | 230       |  |
| vapor)   | 6       | 27.84              | 230       |  |
|          | 7       | 31.40              | 230       |  |
|          | 8       | 51.26              | 230       |  |
| V-band   | 9       | 52.28              | 230       |  |
|          | 10      | 53.86              | 230       |  |
|          | 11      | 54.94              | 230       |  |
| (oxygen) | 12      | 56.66              | 600       |  |
|          | 13      | 57.30              | 1000      |  |
|          | 14      | 58.00              | 2000      |  |

It is vital to perform instrument calibrations, in order to determine the relationship between the detector voltage  $U_{det}$  measured and the received radiation intensity at the antenna, which is expressed as black-body equivalent brightness temperature  $T_B$ . This relationship is described by Eq. 1 (see also Maschwitz et al., 2013):

$$U_{det} = G \left( T_{SC} + T_R \right)^{\alpha} \tag{1}$$

In this equation, the detected signal consists of the receiver noise temperature  $T_R$  and the scene temperature  $T_{Sc}$  received from outside the system. The proportionality factor G (referred to as gain) represents the

amplification of the system, while  $\alpha$  is a factor accounting for potential nonlinear behavior in the receiver components. If the observed source is a blackbody, the physical temperature of the source corresponds directly to its brightness temperature. During the calibration process, two reference blackbody targets with known temperatures ( $T_{Sc}$ ) are used as inputs, while the parameters  $T_R$ , G, and  $\alpha$  in Eq. 1 are determined. In addition, a built-in noise diode is used during calibration, adding a further term to the calibration equation (Maschwitz et al., 2013). Regular calibration against blackbodies spanning a range of temperatures – commonly known as hot/cold calibration – is essential to ensure accurate and stable long-term measurements by correcting for instrumental drift and nonlinearity effects. In the lower temperature range, the blackbody target (also called "cold load") is constructed as a container with sidewalls that are transparent for microwave radiation. During the calibration process, the cold load is attached to the MWR and filled with liquid nitrogen (LN2), which has a boiling point of about 77.3 K at sea level. Conversely, the blackbody in the higher temperature range ("hot load") is housed within the MWR. Its temperature slightly surpasses ambient conditions and is also employed for automatic relative calibrations occurring every 5-10 min. For Gen5 HATPROs, the latest cold load targets aim for a calibration error below 0.25 K in all channels, as specified by the manufacturer RPG (RPG-Radiometer Physics GmbH, 2016, 2022b). A more comprehensive description of LN2 calibrations and their utilization is provided in Section 3.2.

RPG also states that the overall absolute accuracy of  $T_B$  measurements should stay below 0.25 K for Gen5 HATPROs as well, with variations across each channel (RPG-Radiometer Physics GmbH, 2022a). This accuracy primarily encompasses various instrument errors, the focal point of this study, including radiometric noise, drifts, calibration repeatability, biases/differences between instruments, and radome degradation. The radome is a weatherproof enclosure that protects the radiometer antenna but is transparent to microwave radiation.

Other radiometer characteristics like antenna beam width and receiver bandwidth (as well as atmospheric propagation) can also have an impact on scanning MWR measurements, depending on the frequency channel and elevation angle (Han & Westwater, 2000; Meunier et al., 2013). However, this study only analyzes zenith, cold-, or hot load observations, for which these characteristics are irrelevant. Additional sources of uncertainty for MWR products like *T* profiles, *IWV*, and *LWP* are due to the radiative transfer model, absorption coefficient uncertainties as well as limited information content.

#### 3. Data and methods








This section outlines the datasets and the essential methods that are employed to obtain MWR instrument uncertainties. Finer details are described when needed in the "Results and discussion" section (Section 4).

## 3.1. Data from HATPRO instruments at MOL-RAO and JOYCE

This study is based on different datasets from observations and experiments at two locations in Germany, Lindenberg and Jülich (see Table 2).

Data for obtaining the repeatability and for biases/differences between instruments originate from a calibration campaign during FESSTVaL 2021 (Field Experiment on Submesoscale Spatio-Temporal Variability

in Lindenberg) (Hohenegger et al., 2023) at the DWD Meteorological Observatory – Richard Aßmann Observatory (MOL–RAO). The calibration campaign, which mainly took place in May and Jun. 2021 (see Section 3.3 for more details), incorporated two co-located Gen5 HATPROs, named DWDHAT and FOGHAT, as well as another Gen2 HATPRO from the University of Hamburg. The latter was only used for reference measurements needed for determining calibration repeatability and was positioned roughly 50 m away from the other two MWRs. Due to the close proximity of all MWRs from May to Aug. 2021, it was possible to compare zenith observations to assess  $T_B$  measurement differences between the DWDHAT and FOGHAT.








Data for obtaining  $T_B$  measurement drifts and jumps via long-term observations over the last seven years (2018–2024) on cold load targets originate from an HATPRO called TOPHAT at the Jülich ObservatorY for Cloud Evolution (JOYCE) (Löhnert et al., 2015). It is worth noting that TOPHAT is significantly older than all other HATPROs used in this study. Originally built in 2009 as an earlier-generation (G2) radiometer, it has since been retrofitted with a 64 Hz rapid noise-switching diode and new receivers, effectively making it a Gen5 HATPRO on paper. However, due to its age and modifications, TOPHAT may not be fully representative of newer state-of-the-art Gen5 HATPROs.

Radiometric noise data from hot load and cold load observations were collected for all Gen5 HATPROs from both the MOL-RAO and JOYCE sites, including data from a new Gen5 HATPRO called JOYHAT which was installed at JOYCE in Jul. 2023.

The radome degradation process is evaluated using data from the DWDHAT. It was operated at MOL-RAO from Apr. 2019 through Apr. 2023 with two significant interruptions: 22 Mar. 2021 to 23 Apr. 2021 and 20 Sep. 2021 to 7 Dec. 2021. During the operation period, several radome changes took place. The case studies we present rely on FESSTVaL data published by Löhnert et al. (2022).

Table 2: Overview of all analyzed Gen5 HATPROs at Lindenberg (MOL-RAO) and Jülich (JOYCE).

| Site    | HATPRO Name | Period                |  |  |
|---------|-------------|-----------------------|--|--|
| MOL-RAO | DWDHAT      | Apr. 2019 - Apr. 2023 |  |  |
|         | FOGHAT      | May 2021 - Aug. 2021  |  |  |
| JOYCE   | ТОРНАТ      | May 2018 – Apr. 2024  |  |  |
|         | JOYHAT      | Jul. 2023 – Apr. 2024 |  |  |

Apart from directly analyzing the  $T_B$  observations for quality control, the RPG software also provides products retrieved from the observations, using e.g.,  $T_B$  and 2 m in-situ temperature, humidity, and pressure). The underlying algorithms (retrievals) RPG uses are neural networks trained on a mix of radiosonde observations and atmospheric model data. The spectral (SPC) retrieval predicts the most likely  $T_B$  spectrum at 109 frequencies between 1 and 104 GHz corresponding to the current atmospheric state. We rely on the SPC retrieved spectra to detect inconsistencies compared to measured  $T_B$  spectra. The SPC-retrieval uses as an input the barometric pressure, the  $T_B$ s from the 14 HATPRO channels, and the values from the sine and cosine of the day of the year. Channel 10 has a sufficient absolute sensitivity to liquid water on the radome, while the SPC retrieval supplies relatively stable predictions of the  $T_B$ s in this channel. Therefore, we use channel 10 (53.86 GHz) for the detection of a wet radome. Differences between observation and SPC retrieval exceeding a defined threshold indicate that the observed  $T_B$ s cannot be matched with a spectrum purely originating from atmospheric emissions.

A wet radome can occur without the internal rain sensor being activated (e.g., due to limited sensitivity). We therefore assessed the rain detection rate of the TOPHAT stationed at JOYCE by comparing it to an independent reference instrument. HATPRO instruments are equipped with a WXT536 weather station from Vaisala. The station detects precipitation with a piezoelectric precipitation sensor (Salmi et al., 2005). The independent reference to the pre-installed rain detector of the TOPHAT is the OTT-Parsivel2, a laser optical disdrometer. It is co-located with the TOPHAT. The Parsivel2 records the rain rate  $\it R$  with a 1 min resolution and a normalized relative error of  $\pm 7.4$  % for liquid precipitation (Park et al., 2017).

#### 3.2. Absolute calibration standard procedure









In order to properly determine many of the MWR instrument errors discussed in Section 4, absolute calibrations with LN2 must be performed (Maschwitz et al., 2013; Küchler et al., 2016). This is done with the help of a cold load target, a container that gets attached and filled with LN2 and acts as a blackbody with a known temperature (77.3 K at 1013.25 hPa). This study worked with the newest cold load targets named PT-V1 and PT-V2. They were developed to eliminate reflections, standing waves, entrainment of oxygen, and rapid evaporation of LN2, all of which had been persistent errors of older cold load targets (Pospichal et al., 2012; Maschwitz et al., 2013; Küchler et al., 2016). The difference between the PT-V1 and PT-V2 targets is in how they polarize the signal. The PT-V1 has two transparent windows for microwaves, one for the K-band and one for the V-band, while the PT-V2 only has one window for both bands. This makes the calibration process easier and faster, as the PT-V2 does not need to be turned around during calibration. According to the manufacturer RPG, both are supposed to have the same calibration error of below 0.25 K in all channels (RPG-Radiometer Physics GmbH, 2016, 2022b). The primary source of errors for an LN2 calibration is condensation of atmospheric water vapor on the radome or the cold load target window, especially during cold and/or rainy weather with high relative humidity above 85 %. Days with such conditions should be avoided for calibration.

The procedure for an LN2 calibration which has been implemented in this study – later referred to as the calibration standard procedure – is as follows: (1) Observe  $T_{\rm B}$ s for 3–5 min on the filled cold load target before the calibration (for both K- and V-band). (2) Run the calibration program in the RPG software with an integration time of 100 s (for both K- and V-band). (3) Observe  $T_{\rm B}$ s for 3–5 min on the cold load target after the calibration (for both K- and V-band). (4) Observe the internal hot load target for 3–5 min.

This procedure is, in principle, applicable to all types of MWRs; the critical factor is the  $T_B$  observations on the cold load before and after a calibration so that those can be compared to detect drifts or jumps in the system calibration. Sticking to this procedure will, therefore, help the operator determine drifts/jumps of the MWR, as well as noise levels (more details on how this works in Sections 4.1 and 4.2.1). Currently, the software from RPG – with which the calibration is carried out – does not support an automatic approach to this procedure. That is why every operator should execute this procedure for each LN2 calibration to yield comparable calibration results throughout an MWR network.

Supplementary information on how to properly calibrate HATPROs can be found in a microwave radiometer calibration document<sup>6</sup> issued by ACTRIS.

<sup>&</sup>lt;sup>6</sup> https://www.actris.eu/sites/default/files/inline-files/CCRES\_MWR\_calibration\_v1\_20231025-1\_3.pdf (last access: 16 July 2024)

# 3.3. Calibration campaign








The calibration campaign that took place at MOL-RAO in May and Jun. 2021 during FESSTVaL and featured three HATPROs in close proximity to determine calibration repeatability and measurement differences between instruments.

Here are the steps of the calibration campaign at MOL–RAO listed in detail: (1) The two co-located Gen5 HATPROs DWDHAT and FOGHAT were calibrated three times each in a row (three calibration rounds) on 5 May and 6 May 2021 employing the calibration standard procedure explained in Section 3.2. (2) Zenith measurements of the atmosphere on both DWDHAT and FOGHAT were conducted between and after calibrations. (3) A nearby third HATPRO called HAMHAT was calibrated once at the beginning of the campaign and then always measured in zenith; it was only used as a reference to determine the repeatability of DWDHAT and FOGHAT. (4) Additional calibrations of DWDGHAT and FOGHAT were carried out later during the summer on 11 May and 15 Jun. to understand measurement differences between these instruments better.

#### 3.4. OmB statistics

Comparing observed  $T_B$  to simulated values (e.g., from an NWP forecast), referred to as observation minus background (OmB), has shown to be valuable for  $T_B$  bias correction (Martinet et al., 2017, 2020) and MWR data assimilation (Vural et al., 2024). Additionally, OmB can be used as a statistical quality monitoring tool to detect faulty absolute calibrations (De Angelis et al., 2017) due to error sources described in Section 3.2, which can cause large jumps in OmB differences. In a network configuration, this tool can inform instrument operators about the need to perform a calibration and improve data quality. In order to derive background  $T_B$ , a microwave radiative transfer (RT) model is needed to compute radiances (expressed as  $T_B$ ) corresponding to the atmospheric state. As input, RT models require thermodynamic profiles (namely atmospheric pressure, temperature, and humidity), which can be extracted from the NWP models or radiosonde profiles.

Here, a framework for OmB monitoring to detect large jumps in  $T_B$  differences, suitable for MWR network application, is presented. Results for TOPHAT, which is part of the ACTRIS network, are shown in Section 4.2.2. Observational data are processed and quality controlled by the software package MWRpy (Marke et al., 2024), which is already implemented in the operational processing framework of ACTRIS. As the background, profiles from the Integrated Forecasting System (IFS) model of the European Centre for Medium-Range Weather Forecasting (ECMWF), are used as input. The forecasts are initialized at 12 UTC on the previous day.  $T_B$ s are simulated with a non-scattering microwave RT model based on Rosenkranz (2017), which considers absorption parameters for oxygen, water vapor, and nitrogen for clear-air atmospheres and accounts for the speed dependence of 22 and 118 GHz line shapes (Rosenkranz & Cimini, 2019) in an updated version from 2024. To avoid high uncertainties coming from the model and absorption of cloud liquid water, only scenes without liquid water clouds are considered. Suitable liquid-cloud-free cases are selected by applying a threshold of 0.5 K to the  $T_B$  standard deviation of the 31.40 GHz channel within 1 h around the model output time, similar to the method in De Angelis et al. (2017). In addition, the model-derived LWPshould not exceed 1 g m<sup>-2</sup>, allowing more cases to pass the selection criteria without substantially impacting the RT results. The OmB analysis then comprises the model background and observed

hourly mean  $T_B$  differences. It should be noted that OmB differences can be attributed to instrument uncertainties, as well as model errors (especially biases in water vapor) from the NWP and RT models, as described in Carminati et al. (2019) and Cimini et al. (2018), respectively.

#### 4. Results and discussion








This section focuses on five key sources of instrument uncertainty in Gen5 HATPRO microwave radiometers: (1) radiometric noise, (2) long-term drifts or jumps, (3) calibration repeatability, (4) systematic differences between instruments, and (5) radome degradation. These uncertainties are discussed in detail in the following sections in this order. The analysis is based on observed brightness temperatures ( $T_B$ ) rather than retrieved atmospheric products, and all findings are derived from Gen5 HATPRO instruments. A conceptual overview of these instrument uncertainties is illustrated in Figure 1.

Radiometric noise represents a fundamental source of instrument uncertainty, as it is intrinsic to each instrument and cannot be reduced or adjusted by the operator. It is important to assess the noise levels of the brightness temperatures, as they determine accuracy and information content of retrieved atmospheric products. Additionally, receiver components of MWRs undergo gradual changes over time and can influence the performance. These slow shifts are here categorized as drifts, while abrupt changes can be described as jumps. Periodic absolute calibrations are designed to correct for these drifts and jumps, though they introduce their own uncertainties. Monitoring OmB statistics can help identify faulty calibrations and determine when a new absolute calibration is necessary. Therefore, being aware of the uncertainties across repeated calibrations is essential. Furthermore, systematic differences between two co-located instruments have been observed. These differences often tend to be missed by operators, as MWRs are rarely installed side by side at the same location. However, it is crucial to recognize that such biases exist. Finally, monitoring and knowing the effects of radome degradation provides an additional layer of quality control, particularly for improving measurement accuracy after rain events.

#### 4.1. Radiometric noise

Radiometric noise refers to random fluctuations in the signal detected by the MWR. It stems from various sources, including electronic components, thermal effects, and inherent characteristics of the measurement system. Radiometric noise can be determined via noise covariance matrices which provide single channel noise levels on their diagonal and correlated noise in the off-diagonal elements (Figure 2). These covariance matrices are available from the HATPRO software after each absolute calibration and are determined both from cold load and hot load target observations. The square root of the diagonal elements within the covariance matrix corresponds to the  $T_B$  standard deviation, which is used to indicate the noise level of a channel in this study. The radiometric noise can also be determined without performing an LN2 calibration by calculating the  $T_B$  variance when observing the internal hot load target. Noise analysis has been conducted on all Gen5 HATPRO instruments for cold and hot load targets, usually both before and after LN2 calibrations, encompassing a minimum of three calibration instances with various 3–5 min observation intervals for each Gen5 instrument.

Figure 2: Typical  $T_B$  covariance matrix for FOGHAT (Gen5 HATPRO) with a 5 min observation time on the hot load target (1 s integration time). The square root of the elements on the diagonal yields the noise values for each channel.






Figure 2 shows a typical hot load covariance matrix from the FOGHAT instrument when it was stationed at MOL–RAO. K-band channels produce less noise than V-band channels, primarily due to less thermal noise. However, different receiver techs between K- and V-band, like different circuit components, antenna specifications, and differently applied system noise temperatures (see radiometer formula in Eq. 2 and 3), also lead to this behavior. To mitigate higher noise levels in the upper V-band channels, larger frequency bandwidths are employed (see Table 1).

Fluctuations in the internal hot load temperature can exhibit notable temporal shifts even within brief intervals of 5 min. Consequently, a linear detrending procedure is applied to the measured  $T_B$  values on the hot load prior to variance and covariance calculations to account for this aspect. Additionally, adjustments are made to eliminate abrupt  $T_B$  changes induced by periodic relative hot load calibrations occurring every few minutes. Unlike the hot load temperatures, cold load temperatures remain constant during absolute calibrations, averting the need for similar detrending procedures.

Generally, noise levels are lower for low and higher for high (brightness) temperatures and decrease with higher channel bandwidths and longer integration times. This corresponds with the radiometer formula, which describes the sensitivity  $\delta T_B$  of an ideal radiometer (Ulaby & Long, 2014):

$$\delta T_B = \frac{T_{Sc} + T_R}{\sqrt{\Delta \nu \cdot \tau}}.$$
 (2)

The sum  $T_{Sc}+T_R$  describes the total system noise temperature. It consists of the  $T_B$ s of an observed scene  $T_{Sc}$  and the noise  $T_R$  from radiometer receiver components.  $\Delta \nu$  describes the frequency bandwidth, and  $\tau$  the integration time of a scene. Noise levels in this study have been detected by using every  $\tau\approx 1\,\mathrm{s}$  measurement of the HATPROs. Reducing the sample size by calculating a mean over more than  $1\,\mathrm{s}$  (e.g.,  $2\,\mathrm{s}$  or  $5\,\mathrm{s}$ ) will decrease the noise levels by  $1/\sqrt{\tau}$  per the radiometer formula. Gen5 HATPROs add noise into the observation with noise diodes  $(T_N)$  via continuous rapid noise-switching. The manufacturer RPG states that this cancels receiver instabilities, i.e., non-white noise contributions. Hence, Gen5 HATPROs obey this extended noise-adding radiometer formula (Ulaby & Long, 2014) with a theoretical sensitivity of:

$$\delta T_B = 2 \cdot \frac{T_{Sc} + T_R}{\sqrt{\Delta v \cdot \tau}} \cdot \left( 1 + \frac{T_{Sc} + T_R}{T_N} \right). \tag{3}$$

Typical values for  $T_N$  are ~1000 K for the K-band and ~2000 K for the V-band, values for  $T_R$  can range from ~300 K to ~600 K for the K- and V-band, respectively (Küchler et al., 2016; RPG-Radiometer Physics GmbH, 2022a). Considering these factors, it is possible to calculate theoretical noise levels for cold load and hot load targets (see thick gray lines in Figure 3) with typical values for  $T_R$  and  $T_N$  as described above (with a  $\pm 20$  K range for  $T_R$ , a  $\pm 100$  K range for  $T_N$ , and where  $T_{SC}$  for the cold load is exactly 77.3 K, while  $T_{SC}$  for the hot load ranges from 280–310 K).

Figure 3: Mean radiometric noise and standard deviation for every channel for the cold load (a) and hot load (b) observations of four different HATPRO instruments (DWDHAT, FOGHAT, TOPHAT, and JOYHAT). Integration time is 1 s. The N indicates the number of 3–5 min observation intervals on the cold or hot load retrieved from before and after calibration events, for which CoVar matrix diagonals and their square roots have been calculated. The thick green lines show theoretical noise values derived from the noise-adding radiometer formula (Eq. 3) with typical system noise and noise diode temperature ranges.

Mean noise levels for Gen5 HATPROs are shown in Figure 3. The values calculated for 3 min cold load target observations with 1 s integration time are usually well below 0.11 K for the K-band, while noise for the lower V-band channels can reach up to 0.24 K (0.13 K without TOPHAT). For 5 min hot load target observations with 1 s integration time, mean noise levels can reach 0.17 K for the K-band and up to 0.32 K (0.18 K without TOPHAT) for the lower V-band channels. It has to be noted that TOPHAT usually shows the highest mean noise levels and variabilities compared to the other Gen5 radiometers, especially within the V-band. This is most probably due to TOPHAT originally being an older and noisier Gen2 instrument with the now retrofitted Gen5 noise-switching diode. This difference in noise levels showcases that it is important to determine noise levels for each instrument individually.

Analyzing calibration log data from TOPHAT and FOGHAT reveal that system noise temperatures  $(T_{Sc} + T_R)$  within the V-band for TOPHAT are around 200–300 K higher, and noise diode temperatures  $(T_N)$  up to 300 K lower compared to FOGHAT, leading to an overall noisier behavior of TOPHAT (see Figure 3). Additionally, TOPHAT has exhibited a drift in system noise of approximately +5.0 K per year within the V-band since 2018 (channel 13 even has a drift of more than +12.0 K), which may hint at an aging V-band receiver component (note that TOPHAT is the oldest HATPRO in this study). However, this increase in system noise

temperature does not noticeably affect overall noise levels yet, as it is too tiny (the impact would be around +0.001 K per year according to Eq. 3). The other HATPROs, at the moment, do not exhibit significant drifts in system noise temperature. All analyzed HATPROs, except for TOPHAT, adhere to these theoretical noise levels fairly well.

Overall, radiometric noise is instrument-specific, random, and operator-independent, while calibrations do not influence general noise levels. In practical scenarios, noise associated with cold loads is better suited for K-band measurements, and noise associated with hot loads is better suited for V-band measurements, as the atmospheric temperatures observed in the different bands usually lie close to those respective temperature regions of the different loads.

#### 4.2. Drifts and jumps in measurements






Analyzing the instrument's stability over time is crucial for determining the interval for absolute calibrations. Drifts, in general, represent the tendency of  $T_B$  measurements to slowly diverge over time from an initial reference level or "truth" after a calibration due to very subtle changes in receiver components. Calibrations on the internal hot load target are usually performed automatically every 5 min, while calibrations with LN2 on the cold load target are typically performed every few months. After such a calibration, slowly growing deviations in  $T_B$  measurements should theoretically reset and jump back to their initial reference level. We can measure the magnitude of these jumps via observing  $T_B$  differences at the cold load target immediately before and after an LN2 calibration. With these observed cold load jumps we can infer the drift (see Section 4.2.1). The term 'drift' here describes the total deviation in  $T_B$  measurements between two absolute calibrations with the assumption that no other random jumps occur between calibrations. Since it is often uncertain whether such random jumps occur, it is generally more accurate to use the broader term 'jump' rather than 'drift', although the two are largely interchangeable in this study.

As already mentioned in Section 4.1, there are also drifts in parameters like system noise or noise diode temperatures. However, these drifts are usually not directly connected to the observed drifts/jumps in  $T_B$  measurements, as they are too small to have a noticeable impact.

# 4.2.1. Observed drifts/jumps at the cold load






In order to determine long-term drifts, we analyze LN2 calibrations and their impact on  $T_{\rm BS}$ . Figure 4 shows the observed 3 min mean  $T_{\rm B}$  differences when looking at the LN2 target before and after calibration for the TOPHAT at JOYCE from May 2018 to Apr. 2024. Channel 6 is known to have problems with this particular instrument and is, therefore, neither representative nor discussed within this drift analysis. The values from the two calibration events in Aug. 2019 were also discarded, as the time between these calibrations was too short (a few weeks to a few hours) to be considered for drifts and are therefore not part of this discussion.

Figure 4: Difference in  $T_B$ s (3 min mean before minus 3 min mean after a calibration) per channel for the K-band (a) and the V-band (b) for the TOPHAT at JOYCE after absolute calibrations. Data from the calibration event in Jul. 2022 is missing.

Focusing on the K-band channels first, it seems obvious that the calibration event in Sep. 2022 is an outlier. Without that event, observed cold load jumps range from  $\pm 0.4$  K and are most of the time for most channels lower than  $\pm 0.3$  K, while showing no tendency for a particular direction. Unfortunately, there are no observation data from the calibration event in Jul. 2022, which is the event before Sep. 2022. Initially, the Jul. 2022 calibration might appear faulty due to a large jump in OmB at that time (see Figure 5a and Section 4.2.2 for more details on OmB). However, OmB briefly returns to near-zero values immediately after the calibration, suggesting that the procedure itself was reasonably accurate. Still, OmB shows strong deviations both before and several days after the Jul. 2022 event – exceeding  $\pm 2$  K – indicating an underlying issue with the instrument, though not necessarily with the calibration itself. In contrast, the large jump seen during the Sep. 2022 calibration seems to effectively correct the prior offset and is therefore likely a valid response to the preceding deviation. The cause of the observed OmB fluctuations around the Jul. 2022 event remains unclear. A possible explanation can be that high outside temperatures during that time combined with insufficient cooling led to unusually high receiver instabilities.

Shifting the focus to the V-band reveals that the observed cold load jumps tend to be negative, while the overall range is much broader than in the K-band, ranging from -1.5 to -2.8 K in the worst case to  $\pm 0.2$  K in the best case. Large cold load jumps are not likely be caused by calibration errors, as none of those differences are followed up with an appropriate compensation at the next calibration event. It is also important

to note that the duration between LN2 calibrations does not seem to have a discernible effect on the observed cold load jumps. While long-term drifts are expected to increase with the duration between LN2 calibrations, this does not hold true in many cases. Overall, this data suggests that there are no intelligible and discernible drift patterns, as the cold load jumps seem to change randomly after each LN2 calibration, with no clear explanation of what causes substantial jumps/drifts, at least in the V-band. This limits the ability to reliably quantify calibration-induced jumps/drifts. For a better understanding of drifts, on the one hand, more calibration events from also other instruments with long enough time periods are necessary. On the other hand, OmB statistics could help to better determine and evaluate long-term drifts. Refer to Section 4.2.2 (and Section 3.4) for further details.

The operator cannot directly influence drifts, but the higher the frequency of LN2 calibrations, the lower the theoretical risk of large drifts. Hence, drifts/jumps should be determined after every calibration, and calibrations should be done frequently. The suggestion would be at least every 6 months but not more often than every 2 to 3 months.

In general, larger jumps/drifts observed on the cold load target are more influential to K-band measurements than to V-band measurements, as K-band measurements during normal operation are usually much closer to the temperature regime of the cold load. In contrast, V-band measurements (except for channels 8 and 9) are usually much closer to the temperature regime of the internal hot load, at which calibrations occur every 5 min.

#### 4.2.2. Determining drifts and jumps with 0mB statistics







One aspect of monitoring instrument stability of MWRs is the analysis of Observation minus Background (OmB) data, aimed at discerning long-term drifts that could inform the operator when to perform LN2 calibrations. However, interpreting OmB data presents inherent complexities.

In Figure 5, OmB daily mean  $T_B$ s of the K-band and V-band for clear-sky cases are presented for the whole year 2022. The dashed lines show days of calibration. Days are only analyzed when there were at least 3 hours clear-sky in both observation and model. The significant jump in OmB after the calibration in mid-Jul. 2022 was initially thought to be due to a faulty calibration at the K-band receiver but is more likely due to some unusual high receiver instability before and after that calibration (see Section 4.2.1). On the other hand, no significant drifts/jumps can be detected in the V-band and absolute calibrations only show a minor impact due to the higher  $T_B$  values compared to the LN2 temperature. Unfortunately, we cannot compare this to actual cold load observations, as we are missing these observations for this particular instance. The other significant jump in the opposite direction right after the calibration event in Sep. 2022 can be interpreted as a correction of the preceding calibration event from Jul. 2022 and what followed after. This would also explain the significant jump in cold load observations from this event, as seen in Figure 4a.

One notable challenge arises from the variability in OmB fluctuations and random jumps, often exceeding the observed cold load jumps. While efforts to track drifts using moving monthly means during post or intercalibration periods are undertaken, model uncertainties and inherent fluctuations frequently obscure clear interpretations. This challenge is exacerbated when attempting to compare jumps after calibrations between the instrument's actual cold load readings and OmB data. When significant jumps within the K-band are observed in both the instrument's cold load readings and OmB data before and after calibrations, they

align closely. However, when only small changes are detected after calibrations, these smaller jumps are often overshadowed by the inherent fluctuations in OmB, making accurate comparisons challenging.

The underlying factors driving high OmB fluctuations, jumps, and biases in the K-band are multifaceted. Potential contributors include model uncertainties with water vapor, atmospheric phenomena such as cloud interference, and challenges associated with clear-sky filtering. In contrast to the K-band, observations within the V-band exhibit fewer fluctuations and almost no visible jumps. This is because most V-band channels usually measure in much warmer temperature regimes and are, therefore, not very susceptible to changes in cold load temperature regimes. When there are discernible jumps in the V-band, they are typically attributable to calibration anomalies.

Figure 5: Daily means of OmB brightness temperature differences for TOPHAT at JOYCE during 2022, (a) for K-band (Receiver 1) and (b) V-band (Receiver 2). The dashed lines indicate times of LN2 calibrations. Observations and background are clear-sky only.

In order to assess whether the jumps during absolute calibrations (Figure 4) can be detected by OmB monitoring, we compared for each channel the 14-day mean OmB differences (clear-sky only) before an absolute calibration, alongside the observed jumps in cold load readings from May 2018 to Apr. 2024 (Figure 6a). In an ideal scenario with perfect calibrations, the OmB deviations immediately preceding an absolute calibration would exhibit the maximum drift since the last calibration, aligning perfectly with the observed cold load jumps. This would result in all data points falling along the  $45^{\circ}$  line in the plot. However, Figure 6 reveals that this is often not the case. The x-axis conveys the same information as in Figure 4a. Excluding the faulty channel 6 and the outlier calibration event in Sep. 2022, the measured differences in  $T_{\rm BS}$  at the cold load remain within  $\pm 0.4$  K. However, the OmB data frequently shows much larger deviations, reaching up to -1.8 K, with channel 1 exhibiting the highest discrepancies. This is likely due to the IFS model underestimating atmospheric water vapor, to which channel 1 is particularly sensitive. One exception where OmB closely aligns with the measured cold load jumps is the outlier calibration event of Sep. 2022, which shows similarly large deviations in both datasets. When averaging over all K-band channels (Figure

6b), the OmB differences still remain between +0.5 K and -1.1 K. A further possible explanation for these observed discrepancies is the LN2 calibration uncertainty of approximately 0.25 K, which is determined at the 77.3 K boiling point of LN2, whereas atmospheric brightness temperatures in the K-band can be significantly lower (down to around 10-15 K). Using the findings of Küchler et al. (2016) as a basis, linearly extrapolating the uncertainty of 0.25 K at 77.3 K down to 15 K results in an uncertainty of approximately 0.35 K. This value could be even higher, as the receiver usually exhibits a slightly non-linear response in practice.




Our analysis shows that discerning long-term drifts with OmB statistics is challenging. The variability and uncertainties of OmB are too high to properly monitor drifts/jumps of under 1 K. However, OmB statistics are a good indicator for faulty calibrations or faulty receiver channels with deviations that exceed 1 K. From the data we have analyzed, we suggest that if the operator encounters daily mean OmB values of > |2 K| in three or more K-band channels for longer than a week (in sufficient clear-sky scenarios, e.g., at least 3 h per day), an LN2 calibration is needed.

Figure 6: Comparison of observation minus background (OmB) T<sub>B</sub>s as a 14-day mean (clear-sky only) before an absolute calibration with the observed T<sub>B</sub> differences at the cold load during absolute calibrations

for TOPHAT, (a) for channels 1–7 from the K-band separately, and (b) the mean over all channels. Data from 13 absolute calibrations which took place between May 2018 and Apr. 2024 are used.

#### 4.3. Calibration repeatability






The calibration repeatability quantifies the capability to perform multiple subsequent calibrations (under the same environmental conditions and on very short time scales with negligible drift) with the same quality with reference to stable, independent  $T_B$  observations. The repeatability can, therefore, be determined via changes to zenith reference measurements after two immediate consecutive calibrations. This means that one MWR serves as a reference instrument, calibrated only once for this study at the beginning of the calibration campaign at MOL–RAO on 5 May 2021 and to which mean zenith measurements of other MWRs after several LN2 calibrations are compared. The change in difference to this reference instrument after a calibration determines the repeatability, shown in Figure 7. In this study, we analyzed the repeatability of the two state-of-the-art Gen5 HATPROs, DWDHAT and FOGHAT, with the help of zenith measurements from another HATPRO (HAMHAT) as a reference. The results yield an absolute repeatability of well below 0.13 K in the K-band and 0.16 K in the V-band for both HATPROs compared to the reference radiometer for 30 min mean zenith measurement after each calibration. The zenith measurements took place on 6 May 2021 during fair weather (8–10°C and a relative humidity of 50–55%) with minimal cloud cover.

Figure 7: Calibration repeatability per channel for the Gen5 HATPROs DWDHAT and FOGHAT. Delta TB denotes the changes to zenith reference measurements after two immediate consecutive LN2 calibrations. The bars show the mean T<sub>B</sub> difference for 30 min zenith observations for DWDHAT (green) and FOGHAT (blue) with respect to HAMHAT. Measurements took place on 6 May 2021.

The operator's dedication to high-quality standards during the calibration can influence the repeatability. Still, it can only be determined when at least two MWRs are simultaneously on the same site. Meaningful values can be best achieved with the same conditions for each calibration (same day, weather, target, etc.). Note that within this study, the repeatability was only determined once for each Gen5 instrument during FESSTVaL at MOL-RAO.

## 4.4. Biases/measurement differences between instruments

Even well-calibrated Gen5 HATPROs with typical random noise values can show systematic differences in observed sky  $T_{\rm BS}$  between two co-located instruments. These differences are referred to as biases in this study and can be assessed through zenith measurements between two MWRs over a specific time period. Such an analysis is only feasible when both radiometers operate at the same location for at least two weeks, as ideally, clear-sky conditions are required. It is important to note that the following results are specific to this particular instrument pair and cannot be generalized to all HATPROs.

Figure 8 presents measurement differences between the two instruments, DWDHAT and FOGHAT, at MOL–RAO during FESSTVaL, with five absolute calibrations performed between May and Aug. 2021. The applied clear-sky filtering method is based on channel 7 (31.40 GHz) zenith observations, selecting cases where the 60 min standard deviation remains below 0.5 K. The analysis within the K-band reveals biases clustered around zero between different calibration periods, whereas the V-band consistently exhibits a negative bias across all channels, days, and calibrations. Absolute mean biases during clear-sky periods from May to Aug. 2021 (Figure 8a) reach up to 0.15 K in the K-band and 0.58 K in the V-band, with channel 8 showing the highest mean bias and variability. Figure 8b illustrates a time series of the rolling 60 min mean for both K-band and V-band biases during the same period, confirming that V-band channels exhibit a higher bias than K-band channels. On average, the bias difference between the two bands is approximately 0.38 K.

A noteworthy observation is an oscillating pattern of about 0.2 K in the mean K- and V-band time series (Figure 8b). This pattern follows a diurnal cycle, and thus suggests a possible temperature dependence of certain instrument components. Further investigations are required to determine the extent and implications of these temperature dependencies and whether they also influence instrument differences in other Gen5 HATPROs.

As previously mentioned, these findings should not be considered generally applicable, as they are based on a single instrument pair over a limited time period and may vary among other Gen5 HATPROs. Nonetheless, we consider it important to include these results to highlight that even state-of-the-art HATPROs, calibrated meticulously using identical procedures, may still yield slightly different measurements under the same conditions. Biases are not readily detectable by operators unless at least two MWRs are deployed at the same location. As side-by-side intercomparisons are rarely feasible in operational networks, high-quality LN2 calibrations on all MWRs remain essential to prevent additional measurement errors.

Figure 8: Zenith  $T_B$  difference means of DWDHAT minus FOGHAT during clear-sky periods. The comparison includes continuous observations between May and Aug. 2021. (a) Mean and standard deviation for the whole period per channel. (b) Time series of mean K-band and V-band deviations over time. Data are averaged over one hour.

#### 4.5. Impact of liquid water on radome and radome degradation

The radome of an MWR is the weatherproof enclosure that protects the antenna from the external environment. It is transparent to microwave radiation and needs to be kept clean of dirt, water, and ice accumulation to avoid influences on the measurements. Thus, mechanisms need to be in place to mitigate the effect of disturbances, so that reasonable atmospheric observations can resume as quickly as possible, especially after rain or icing rain events.

#### 4.5.1. Negative effects of a wet radome



A water layer on the radome surface reflects and absorbs radiation in the K-band and V-band and emits radiation according to the water temperature. The magnitude of this effect increases with the thickness of the water film.

As the MWR community has been aware for a long time of biases induced by a wet radome, some mechanisms are in place to prevent the unnoticed collection and use of biased data. The most prominent are introduced in the following. The radome is coated with a hydrophobic layer, which hinders water accumulation. The manufacturer RPG recommends renewing the radome at least every 6 months or sooner if weathering effects are visible. Combined with a heater-blower system, which is activated when rain is detected

and a relative humidity threshold is exceeded, the accumulation of water on the radome is reduced. The pre-installed precipitation sensor supplies a rain flag that gives an indication whether the microwave observations might be affected by a wet radome. Evaluations of MWR observations and retrievals showed that observations at 75° elevation or lower significantly reduce bias and RMS of the resulting retrievals during precipitation events (Ware et al., 2013; Xu et al., 2014; Foth et al., 2024). In addition to these pre-installed mechanisms, several good practice and technical monitoring recommendations were published during the COST action ES0702 EG-CLIMET<sup>7</sup>.

We have identified two potential weak spots in the mitigation mechanisms for a wet radome described above: (1) The precipitation sensor may fail to detect relevant rainfall, and (2) the radome may remain wet after the end of the rainfall, especially when the hydrophobic coating has already degraded. In the following Section 4.5.2, we look into the agreement of rain detection rates of the pre-installed rain sensor and an optical disdrometer, and in Section 4.5.3, we evaluate an additional mitigation strategy in how to deal with a wet radome.

#### 4.5.2. Performance of the piezoelectric precipitation sensor







The pre-installed precipitation sensors on HATPROs may not be sensitive enough for detecting rain collecting on the radome. That is why we assess the performance of the pre-installed Vaisala precipitation sensor mounted to the TOPHAT at JOYCE by comparing it to a co-located Parsivel2 disdrometer. The Parsivel2 is an independent and more accurate sensor than the Vaisala one with a completely different detection scheme and can therefore serve as an independent reference. The detection rate of rain of the pre-installed Vaisala sensor compared to the Parsivel2 is displayed in Figure 9. It ranges between 92 % and 97 % and increases for rain rates  $R \ge 2$  mm h<sup>-1</sup>. These detection rates are sufficient to detect rain and rule out the Vaisala precipitation sensor from being a source of error in our study. However, the data at hand is insufficient to identify a possible seasonality in the detection rates.

Figure 9: Histogram of rain rate R (mm  $h^{-1}$ ) between 1 mm  $h^{-1}$  and 5 mm  $h^{-1}$  split into cases missed (orange) and detected (green) by the Vaisala precipitation sensor on the TOPHAT at JOYCE. The relative detection rate is indicated on the bars. The displayed data was collected over four years, starting from 1 Jan. 2019, with a 1 min resolution.

<sup>7</sup> http://cfa.aquila.infn.it/wiki.eg-climet.org/index.php5/MWR\_Technical\_Implementation (last access: 8 February 2024)

# 4.5.3. Spectral inconsistencies after rain events and radome degradation







When operating an MWR, there are sometimes significantly increased  $T_{\rm B}$ s of several Kelvins immediately after a rain event compared to measurements with a fully dried radome a few minutes later. These biased  $T_{\rm B}$ s do not agree with an expected atmospheric state (see Section 3.1) and lead to unrealistic retrievals of meteorological variables. Nevertheless, these data are not flagged because the rain event was over, and the rain sensor no longer detected precipitation.

Figure 10 shows the behavior of the 53.86 GHz  $T_{\rm B}$ s from the SPC-retrieval relative to the observed  $T_{\rm B}$ s (Observation minus SPC-retrieval or OmSPC) during an intense rain event with a fairly new radome. Slight differences between observation and retrieval only become visible during the most intense rain. This behavior changes significantly if the radome has undergone some aging and degradation. Figure 11 shows OmSPC values during a rain event 4 months after installing a new radome. They reach a maximum of up to 12 K and slowly decrease after the end of the rain event. The increase in OmSPC values coincides with  $T_{\rm B}$ s close to or above ambient temperature in all 14 channels.

According to this behavior, we implemented an additional wet flag (Figure 11 orange shading) for times after detected rain events in which OmSPC in the 53.86 GHz channel is higher than a threshold  $\Delta T_{limit}$ , with  $\Delta T_{limit} = 2~K + mean(OmSPC)_{day}$ . This method was first described by Löffler (2024). The  $\Delta T_{limit}$  is empirically chosen to be well above the intra-day fluctuation of OmSPC values. It changes every day because it depends on the daily mean of OmSPC. Moreover, the flagged time period is extended by an additional buffer time  $(t_{buffer})$  to account for the remaining drying process after the difference has dropped below  $\Delta T_{limit}$  (see Figure 11 purple shading). The  $t_{buffer}$  is defined by  $t_{buffer} = 180~s \cdot \Delta T_{limit}$ . While this is sufficient for the MOL–RAO site, other environmental conditions may require an extension of this time buffer.

Building on the wet flag from the preceding paragraph we now define the "time to dry" as the time between the last detected rain and when OmSPC first drops below  $\Delta T_{limit}$ . In the following we use the "time to dry" as an indicator for the radome condition. It is important to note, that an increased "time to dry" may also have other causes, such as a malfunction of the blower and/or heater.

Figure 10: Observation minus SPC-retrieval (blue curve) of DWDHAT during events with rain on 2 May 2021. The rain rate is shown in orange. Rain events registered by the precipitation sensor are shaded green. The solid blue line indicates the threshold for OmSPC ( $\Delta T_{limit}$ ) on this day.

Figure 11: Observation minus SPC-retrieval (blue curve) of DWDHAT during events with rain on 29 Aug. 2021. The rain rate is shown in orange. Rain events registered by the precipitation sensor are shaded green, while the wet flag is shaded orange, and the time buffer is shaded purple. The solid blue line indicates the threshold for OmSPC ( $\Delta T_{limit}$ ) on this day.

Figure 12a shows the evolution of the "time to dry" for different radomes installed on DWDHAT at MOL-RAO. A steep increase in the "time-to-dry" occurs when the radome age is between 100 and 300 days. We conclude that environmental conditions strongly influence the weathering of the radome. Also, it is not adequate to replace the radome on a fixed schedule every 6 months. For example, the 100 days lifetime of the radome, which we observed starting May 2019, is well below the recommended replacement interval of 6 months provided by the manufacturer, RPG. Monitoring the "time-to-dry" allows for noticing premature weathering of the radome and the degradation of its hygroscopic properties. Figure 12b shows a systematic increase in the "time to dry" during and after summer. This systematic increase indicates that the conditions in summer are especially favorable for the undesired weathering. Intense convective rain and hail play an important role in radome weathering, as does air pollution, but we postulate that the main driver of radome weathering is intense UV radiation, as it also occurs after long episodes with little rain.

The 30 day rolling maximum of the "time to dry" is most instructive because the "time to dry" strongly varies with the length and intensity of a rain event and is also influenced by other parameters such as cloud cover, wind, or relative humidity. It is also worth noting that the described method requires rain events with sufficient intensity and duration. At this stage, we cannot quantify these requirements; however, if there is no rain, the weathering process cannot be tracked.

Figure 12: 30 day rolling maximum of "time to dry" plotted against radome-age (a) and days of the year (b) for DWDHAT. This evolution only contains the worst cases. The radome degradation process is also dependent on the environmental conditions of the site (e.g. precipitation intensity, UV exposure, and air pollution). The more severe the degradation, the longer is the "time to dry".




Our investigations linked to rainwater aggregation on the radome revealed that differences between spectral retrieval and observation (OmSPC) during and after rain events (e.g., see Figure 12) indicate a signal of non-atmospheric origin, i.e., water on the radome. We propose monitoring the "time to dry" as a method for monitoring the radome weathering. This monitoring can increase the availability of unbiased data if a new radome is required ahead of schedule. We recommend replacing the radome once the "time to dry" consistently exceeds 10 min under standard operating conditions.

The Vaisala piezoelectric rain sensor displayed no sign of reduced performance during the analyzed time frame. We conclude that the rain sensor reliably detects rain, even after several years of operation. The described method for flagging and monitoring is only a first step. For many applications (e.g., data assimilation), rain that does not reach the instrument is also not favorable due to scattering and high *LWP*s. Note, that scattering is not accounted for by non-scattering radiative transfer models such as RTTOV-gb or line-by-line models.

#### 5. Recommendations for MWR operators









The following section summarizes the actions that can and should be taken for analyzing instrument uncertainties (see also Table 3) as well as ensuring smooth operation, especially with regard to MWR networks. These recommendations are mainly addressing MWR operators but provide also a basis for monitoring activities in networks. Furthermore, instrument manufacturers get an overview about instrument monitoring requirements.

The operator can and should determine noise levels on a per-channel basis by the noise covariance matrix which is provided for HATPRO MWRs during every absolute calibration. The noise levels cannot be influenced by the operator, but if noise levels increase over time and are continually larger than usual (see Table 4 for specifics), the age limit of the instrument may be reached. The operator should also be aware that gradual changes in system noise and instrument gain can negatively affect instrument performance. Additionally monitoring these parameters is, therefore, beneficial.

Uncertainties concerning absolute calibrations can be influenced by the operator. LN2 calibrations at favorable weather conditions (no precipitation, relative humidity below 85 %) and with a functioning heater/blower prevent the risk of condensation on the cold load target. Strictly following the standard calibration procedure (see Section 3.2) with the newest cold load targets (PT-V1 or PT-V2) should minimize calibration errors and, therefore, also minimize uncertainties associated with calibration repeatability and biases/instrument differences.

To reduce the negative influence of possible long-term drifts, we recommend performing an LN2 calibration at least every 6 months, which is in accordance with the manufacturer's recommendations. For monitoring drifts/jumps, the operator should determine the  $T_B$  differences observed at the cold load before and after each LN2 calibration. Here, one should consider that drifts do not necessarily follow a discernable pattern over time. Monitoring drifts and jumps via OmB is challenging due to its high variability and inherent uncertainties. To be clearly identifiable, drifts or jumps typically need to exceed approximately 2 K. However, this method is effective for detecting faulty LN2 calibrations and malfunctioning receiver channels. A rough estimate of when to calibrate is when daily means of OmB exceed 2 K for longer than a week in three or more K-band channels for clear-sky scenarios which occur for at least 3 h a day.

Comparing observations to co-located radiosonde measurements instead of model outputs would be a more accurate way to monitor drifts/jumps but such radiosonde measurements are usually not available for most operators. If available, however, using co-located radiosondes would be the preferred method. Another possibility of capturing and reducing drifts, especially between LN2 calibrations, could be the use of tip curve calibrations, where the cold target is an independent measurement of the clear sky at different elevation angles under homogeneous conditions. However, the practical implementation of such tip curves is not straightforward and can have uncertainties of up to 0.6 K (Küchler et al., 2016). They were therefore not discussed in this paper. Still, if properly processed and monitored, tip curves can be an accurate way to calibrate K-band channels and help mitigate long-term drifts, especially if LN2 calibrations cannot be performed for an extensive period.

The recommendation from the manufacturer RPG to replace the radome every 6 months is a rough estimate as to when monitoring the "time to dry" is not feasible. We suggest, however, to be alert when the "time to dry" after a rain event exceeds 3 min and to replace the radome at the latest when the "time to dry"

exceeds 10 min. We do not recommend delaying the radome replacement, as degradation may remain unnoticed, e.g., due to lack of rain.

Table 3: Summary of actions an MWR operator can perform to ensure smooth network operation.

|                                     | Action                                                                                                                                                                                      | Recommended<br>Frequency                       | Operational<br>Burden | Priority    |
|-------------------------------------|---------------------------------------------------------------------------------------------------------------------------------------------------------------------------------------------|------------------------------------------------|-----------------------|-------------|
| Noise level determination           | Monitoring noise covariance matrices                                                                                                                                                        | every 6 months                                 | low                   | recommended |
| Absolute<br>calibration<br>with LN2 | Following the standard calibration procedure with the newest cold load targets. Only very large drifts of > 2 K and faulty calibrations can be satisfactorily determined via OmB statistics | at least every<br>6 months                     | high                  | essential   |
| Instrument drifts determination     | Observing cold load $T_B$ s differences before and after LN2 calibrations                                                                                                                   | every 6 months<br>(during LN2<br>calibrations) | high                  | recommended |
| Radome replacement                  | Checking if "time to dry" after rain events exceeds 10 min                                                                                                                                  | at least every<br>6 months                     | medium                | essential   |

# 6. Summary and Outlook



This study analyzed instrument uncertainties of state-of-the-art Gen5 HATPROs based on observations from four different instruments at two locations. Specifically, we examined noise levels on cold and hot loads, jumps after LN2 calibrations as indicators of long-term drifts, calibration repeatability, zenith measurement differences between two instruments, and radome degradation due to weathering. A comprehensive quantitative summary of these instrument uncertainties is provided in Table 4. It is noteworthy that one of the instruments, TOPHAT, was originally built as an earlier-generation radiometer and has been retrofitted to meet Gen5 HATPRO specifications. However, due to its age and modifications, it may not fully represent newer state-of-the-art Gen5 HATPROs, especially when it comes to noise levels and general receiver stability in the V-band. TOPHAT is also the only instrument, at which long-term drifts/jumps were analyzed.

Table 4: Summary of analyzed instrument uncertainties for Gen5 HATPROs. Uncertainties are described as absolute  $T_{B}$ s.

| Type of<br>Uncertainty                                                  | Typical<br>Error Values<br>K-band      | Typical<br>Error Values<br>V-band      | Error determined<br>via                                                                   | Can error<br>be influ-<br>enced by<br>handling? | How to reduce error?                                  | Should error<br>be deter-<br>mined by the<br>operator? |
|-------------------------------------------------------------------------|----------------------------------------|----------------------------------------|-------------------------------------------------------------------------------------------|-------------------------------------------------|-------------------------------------------------------|--------------------------------------------------------|
| Noise Levels (1 s) (3 min cold load and 5 min hot load)                 | ≤ 0.11 K<br>and<br>≤ 0.17 K            | ≤ 0.24 K<br>and<br>≤ 0.32 K            | Standard<br>deviation (from<br>covariance matrix<br>diagonal)                             | no                                              | Not possi-<br>ble;<br>instrument<br>specific          | yes                                                    |
| Jumps after LN2 Calibrations (indicating drift or a faulty calibration) | usually ≤ 0.4 K<br>(up to 1.8 K)       | usually ≤ 1.0 K<br>(up to 2.8 K)       | Differences at<br>cold load before<br>and after a LN2<br>calibration                      | yes                                             | Frequency<br>and quality<br>of<br>calibration         | yes                                                    |
| Calibration<br>Repeatability                                            | ≤ 0.13 K                               | ≤ 0.16 K                               | Changes to zenith reference measurements after two immediate consecutive LN2 calibrations | yes                                             | Quality<br>of<br>calibration                          | no                                                     |
| Mean Biases/<br>Instrument<br>Differences<br>(example)                  | ≤ 0.15 K                               | ≤ 0.58 K                               | Zenith measure-<br>ment differences<br>between two<br>MWRs                                | yes                                             | Quality<br>of<br>calibration                          | no                                                     |
| Radome<br>Degradation                                                   | > 2 K<br>after 10 min<br>"time to dry" | > 2 K<br>after 10 min<br>"time to dry" | Observation minus SPC-retrieval in channel 10                                             | yes                                             | Replace ra-<br>dome when<br>necessary,<br>filter data | yes                                                    |

For better context, we can compare the analyzed instrument uncertainties with previous studies and the manufacturer's information. These studies, however, were mainly focused on calibration uncertainties. For a Gen2 HATPRO, Maschwitz et al. (2013) observed LN2 calibration blackbody uncertainties of  $\pm 0.3$  to  $\pm 1.6$  K, but they were using an older calibration target where the uncertainty is dominated by the reflectivity of the target. For the calibration repeatability, they found values of roughly up to  $0.3 \pm 0.4$  K in the K-band and  $0.2 \pm 0.4$  K in the lower V-band. Küchler et al. (2016) used a newer Gen4 HATPRO and determined a reduced calibration uncertainty of  $\pm 0.5$  K. The HATPROs manufacturer RPG claims there is an absolute  $T_B$  measurement uncertainty of  $\pm 0.25$  K for Gen5 instruments, with  $\pm 0.25$  K as well for the calibration uncertainty with the newest cold load targets. They specify noise levels ranging from 0.10 to 0.25 K and a calibration repeatability of 0.025 K (RPG-Radiometer Physics GmbH, 2016, 2022a, 2022b). Comparing these values with the typical uncertainties found in this study reveals that they do not fully align with all of the manufacturer's claims, especially with respect to long-term drifts as well as biases between instruments. However, within this study drifts have only been analyzed for one instrument, and the biases have been observed during one 3 month period of co-located observations.

A quantification of real MWR observation uncertainty is highly important for data assimilation as well as to accurately retrieve meteorological variables like *IWV* and *LWP*. In the future, a centralized monitoring of instrument calibrations and uncertainties will be established with the implementation of MWR networks, such as ACTRIS and E-PROFILE, including continuous OmB monitoring for each location. These activities will allow assessing specific uncertainties concerning noise and drifts for each network instrument which can then be used for targeted retrieval algorithms. Furthermore, the radome quality monitoring will give individual recommendations for radome changes.

#### Code and data availability





We thank ACTRIS, the Finnish Meteorological Institute, and ECMWF for providing IFS model data for JOYCE from 2022, which are available under O'Connor (2025). HATPRO observations from the same time period and location are available under Pospichal & Löhnert (2025) and have been processed and quality controlled by the software package MWRpy (Marke et al., 2024). HATPRO data used during FESSTVaL at MOL–RAO can be accessed under Löhnert et al. (2022). The code necessary for radome monitoring is available on GitHub under https://github.com/igmk/mwr\_radome (last access: 9 April 2025). Data for all remaining HATPRO measurements and experiments used in this study are available at the Institute for Geophysics and Meteorology of the University of Cologne.

#### **Author contributions**

TB, ML, BP, CK and UL designed this study together. TB evaluated the data from most measurements and experiments, produced most figures, and wrote the manuscript with the help of ML, TM, BP, CK and UL. Section 3.4 and Figure 5 in Section 4.2.2 was provided by TM, while Section 4.5 with all its figures was provided by ML.

#### **Competing interests**

The contact author has declared that none of the authors has any competing interests.

## Disclaimer

Responsibility for the content of the publication lies with the authors.

# Acknowledgements

The authors gratefully thank Annika Schomburg (German Meteorological Service, DWD) and Jasmin Vural (Météo-France) for their support and helpful discussions, as well as Anton Widulla from the University of Cologne for helping to assess MWR data for instrument differences.

The presented research has been made possible through access to two research platforms. (1) JOYCE, Jülich Observatory for Cloud Evolution is operated jointly by University of Cologne and Research Center Jülich (Institute for Climate and Energy Systems 3). JOYCE (www.joyce.cloud, last access: 9 April 2025) is part of the European Research Infrastructure Consortium ACTRIS and is supported by the German Federal Ministry for Education and Research (BMBF) under the grant identifiers 01LK2001G and 01LK2002F. Also, JOYCE is a central measurement infrastructure of CPEX-LAB (Cloud and Precipitation Exploration Laboratory) supported through Geoverbund ABC/J (https://www.geoverbund-abcj.de, last access: 9 April 2025). (2) Access to MOL-RAO, the DWD Meteorological Observatory Lindenberg, was possible during and aligned with the FESSTVaL campaign within the Hans-Ertel-Center for Weather Research (https://www.hans-ertel-zentrum.de, last access: 9 April 2025).

The paper has been motivated by collaborative concepts developed within the EU COST (European Cooperation in Science and Technology) Action CA18235 "PROBE" (Profiling the Boundary Layer on European Scale, https://www.probe-cost.eu, last access: 9 April 2025).

#### Financial support







The presented research has been performed in the TDYN-PRO (Integration of Ground-based Thermodynamic Profilers into the DWD Forecasting System) project within the funding line "Extramurale Forschung" of DWD under the grant number 4819EMF02.

This open-access publication was funded by University of Cologne.

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
