# Peer review of "Instrument uncertainties of network-suitable groundbased microwave radiometers: overview, quantification, and mitigation strategies"

_EGUsphere, 2025_

## Referee Comment (RC1)

Review AMT

**Title**: *Instrument uncertainties of network-suitable ground-based microwave radiometers: overview, quantification, and mitigation strategies*
**Authors**: Tobias Böck et al.
**Journal**: *Atmospheric Measurement Techniques Discussions / EGUsphere*
**DOI**: https://doi.org/10.5194/egusphere-2025-1727

**General comments:**

This manuscript presents a technically rigorous and operationally relevant assessment of instrumental uncertainties associated with state-of-the-art HATPRO-Gen5 microwave radiometers. The study is well-conceived, timely, and aligns with current efforts across initiatives such as ACTRIS, GRUAN, and E-PROFILE to establish standardized, high-quality microwave radiometer (MWR) networks for both research and numerical weather prediction applications.
The work is particularly valuable for its:

- Systematic breakdown of key uncertainty sources (radiometric noise, calibration repeatability, drift/jumps, inter-instrument biases, and radome degradation),
- Use of long-term datasets and controlled calibration campaigns,
- Practical framework for operational traceability (e.g., "time to dry" metric, OmB-based quality control).

**Technical suggestions:**

- **Figure 3, Page 11**
    - The current color choices for representing radiometric noise (red, orange, and purple) are difficult to differentiate. Consider using a more contrasting palette to enhance clarity
- **Page 14, Line 398**
    - The phrase *"this makes it hard to quantify these jumps meaningfully"* could be more formally written as, for example, *"This limits the ability to reliably quantify calibration-induced drifts."*
- **Page 17, Section 4.3**
    - It would be helpful to specify that the repeatability tests were carried out during clear-sky and fair-weather conditions, along with relevant environmental parameters such as temperature and humidity, to support interpretation of the results.
- **Page 18, Line 536**
    - The phrase *"biases are not readily detectable by operators"* could be expanded with a short note that "side-by-side intercomparison are rarely feasible in operational networks" for clarity.
- **Page 23, Figure 12 Caption**
    - Specify whether the "time to dry" metric is site-specific or generalizable. Indicate environmental dependencies (e.g., UV exposure, rain intensity).
- **Page 24, Line 650**
    - Consider rephrasing: *"a radome replacement is necessary if the time to dry is longer than 10 min"* → *"We recommend replacing the radome once the drying time consistently exceeds 10 minutes under standard operating conditions."*
- **Page 24, Section 5**

- The textual list of actions is valuable but could be presented more clearly by providing a summary table. Possible columns for the table: Action, Uncertainty addressed, Recommended frequency, Operational burden (e.g., Low, Medium, High), Priority (Essential / Recommended / Optional).

**Conclusion**

The paper is clearly written, well organized, and meets the high standards expected by Atmospheric Measurement Techniques. With minor revisions as outlined below, the manuscript is fully suitable for publication.

---

## Referee Comment (RC2)

Egusphere-2025-1727: Instrument uncertainties of network-suitable ground- based microwave radiometers: overview, quantification, and mitigation strategies by Böck et al.

The paper examines sources of uncertainties in ground-based microwave radiometers focusing on liquid nitrogen calibration and detection of long-term drifts as well as rain sensor and degradation of the radome. The paper provides recommendation for the use of microwave radiometers in the field. The discussion focuses on one type of radiometers, namely the HATPRO G5 model. The paper is well written and organized and suitable for publication on AMT.

General comment

My main comment relates to some of the assumptions on the LN2 calibration for channels between 20 and 30 GHz. Specifically, the assumption that if the agreement with the estimated cold target temperature (77 K) is in within a certain range, that is also true when the instrument is looking at the sky.

LN2 calibrates the radiometer in a range of temperatures that is mostly outside the range of what is observed. The "cold" target (~77 K) is actually very warm as the measured brightness temperatures can be as low as 10-15 K (or even lower in high latitudes). Therefore, an accurate calibration is achieved between ~77 K and ~290 K (warm target) forcing an extrapolation to the lower temperature that is prone to larger uncertainties.

Therefore, the assumption is section 4.2.2 line 465 that the *"OmB deviations immediately preceding an absolute calibration would exhibit the maximum drift since the last calibration, aligning perfectly with the observed cold load jumps"* is not realistic. The radiometer may achieve a difference of 0.25 K at 77K but may have a larger (or smaller) bias at colder temperatures because of the non-linear response of the receiver. But this would be true even in the case of a perfectly linear receiver (the well-known lever arm error). On the other hand, the range of sky brightness temperatures in the V-band channels is much more likely to be closer to the LN2 calibration cold-warm target. This should be mentioned in line 470 as the discrepancy shown in Fig. 6 may not be entirely due to the model.

[Figure]

The very good (average) agreement shown in Fig. 8 between 2 units frequently calibrated with LN2 is very encouraging, however having so many calibrations is not typical. Chances are that with one calibration every 6 months or less the K-band may be biased.

A second comment relates to the mention of tip curves. Although not the focus of the paper, in my opinion tip curves should be at least mentioned (if not discussed) as a mitigation strategy for long term drifts in between LN2 calibrations. With tip curves the "cold" target is an independent measurement of the sky and is therefore much lower than the LN2 temperature. Although their practical implementation is not straightforward, continuous tip curves, properly processed and monitored are probably the most accurate calibration for the K-band and can well capture drifts and fluctuations in the receiver hardware. Therefore, for completeness of discussion, especially for users non entirely familiar with the instrument, there should be at least a mention of calibration with tip curves as a mitigation strategy for long term drifts between LN2 calibrations.

Minor comments:

Section 4.2.1, Line 400: "*We know, however, that this particular calibration has to be faulty, as indicated by the large jumps in OmB within the K-band after this event (see Figure 5a and Section 4.2.2 for more details on OmB)."* I may be misinterpreting here, but if I look at Fig. 5 where there is the dashed line on July 22 the OmB actually decreases after the calibration suggesting that it was somewhat successful. What am I missing?

Section 4.2.2, Line 476: *"Our analysis shows that discerning long-term drifts with OmB statistics is challenging."* This is certainly true when using model output. If co-located radiosondes are available, uncertainty due to balloon drifts or spectroscopy are generally much smaller than calibration biases and drifts.

---

## Author Comment (AC1)

**Answers to Review #1**

**General comments:**

This manuscript presents a technically rigorous and operationally relevant assessment of instrumental uncertainties associated with state-of-the-art HATPRO-Gen5 microwave radiometers. The study is well-conceived, timely, and aligns with current efforts across initiatives such as ACTRIS, GRUAN, and E-PROFILE to establish standardized, high-quality microwave radiometer (MWR) networks for both research and numerical weather prediction applications.

The work is particularly valuable for its:

• Systematic breakdown of key uncertainty sources (radiometric noise, calibration repeatability, drift/jumps, inter-instrument biases, and radome degradation).

• Use of long-term datasets and controlled calibration campaigns.

• Practical framework for operational traceability (e.g., "time to dry" metric, OmB-based quality control).

> ▪ Thank you for that nice comment. We're glad you appreciate our hard work.

**Technical suggestions:**

• Figure 3, Page 11

The current color choices for representing radiometric noise (red, orange, and purple are difficult to differentiate. Consider using a more contrasting palette to enhance clarity.

> ▪ Done.

• Page 14, Line 398 (you mean line 414)

The phrase "this makes it hard to quantify these jumps meaningfully" could be more formally written as, for example, "This limits the ability to reliably quantify calibration-induced drifts."

> ▪ Changed.

• Page 17, Section 4.3

It would be helpful to specify that the repeatability tests were carried out during clear-sky and fair-weather conditions, along with relevant environmental parameters such as temperature and humidity, to support interpretation of the results.

> ▪ Line 499 in Section 4.3 already states that. We think that the exact temperature and humidity during the test are not important to the overall result as long as the conditions are favorably (fair weather, minimal cloud cover) and stay the same during the whole procedure. Nevertheless, we added the temperature and humidity conditions during that day.

• Page 18, Line 538

The phrase "biases are not readily detectable by operators" could be expanded with a short note that "side-by-side intercomparison are rarely feasible in operational networks" for clarity.

- ▪ Added.

• Page 23, Figure 12 Caption

Specify whether the "time to dry" metric is site-specific or generalizable. Indicate environmental dependencies (e.g., UV exposure, rain intensity).

- ▪ The degradation process is highly dependent on the weather conditions of the site (rain intensity, hail, UV exposure). This is already stated in lines 632 to 634. Nevertheless, we added more context to the caption.
- ▪ Added air pollution to line 634.

• Page 24, Line 650

Consider rephrasing: "a radome replacement is necessary if the time to dry is longer than 10 min" → "We recommend replacing the radome once the drying time consistently exceeds 10 minutes under standard operating conditions."

- ▪ Changed.

• Page 24, Section 5

The textual list of actions is valuable but could be presented more clearly by providing a summary table. Possible columns for the table: Action, Uncertainty addressed, Recommended frequency, Operational burden (e.g., Low, Medium, High), Priority (Essential / Recommended / Optional).

- ▪ Expanded Table 3 on page 24 according to your suggestion.

**Conclusion**

The paper is clearly written, well organized, and meets the high standards expected by Atmospheric Measurement Techniques. With minor revisions as outlined below, the manuscript is fully suitable for publication.

---

## Author Comment (AC2)

**Answers to Review #2**

The paper examines sources of uncertainties in ground-based microwave radiometers focusing on liquid nitrogen calibration and detection of long-term drifts as well as rain sensor and degradation of the radome. The paper provides recommendation for the use of microwave radiometers in the field. The discussion focuses on one type of radiometers, namely the HATPRO G5 model. The paper is well written and organized and suitable for publication on AMT.

- Thanks a lot for this comment.

**General comment**

My main comment relates to some of the assumptions on the LN2 calibration for channels between 20 and 30 GHz. Specifically, the assumption that if the agreement with the estimated cold target temperature (77 K) is in within a certain range, that is also true when the instrument is looking at the sky.

LN2 calibrates the radiometer in a range of temperatures that is mostly outside the range of what is observed. The "cold" target (~77 K) is actually very warm as the measured brightness temperatures can be as low as 10-15 K (or even lower in high latitudes). Therefore, an accurate calibration is achieved between ~77 K and ~290 K (warm target) forcing an extrapolation to the lower temperature that is prone to larger uncertainties.

- That is correct. We added this at lines 475f.

Therefore, the assumption is section 4.2.2 line 465 that the "OmB deviations immediately preceding an absolute calibration would exhibit the maximum drift since the last calibration, aligning perfectly with the observed cold load jumps" is not realistic.

- What we describe here would be an "ideal scenario" with perfect calibrations, which means it's not necessarily realistic per se.
  We added more context here.

The radiometer may achieve a difference of 0.25 K at 77K but may have a larger (or smaller) bias at colder temperatures because of the non-linear response of the receiver. But this would be true even in the case of a perfectly linear receiver (the well-known lever arm error). On the other hand, the range of sky brightness temperatures in the V- band channels is much more likely to be closer to the LN2 calibration cold-warm target. This should be mentioned in line 470 as the discrepancy shown in Fig. 6 may not be entirely due to the model.

- We added (at lines 475f) that the discrepancies shown in Fig. 6 can also be due to a higher calibration uncertainty at very low temperatures of 10-15 K. Using Küchler et al. (2016) as a basis, linearly extrapolating the uncertainty of 0.25 K at 77.3 K down to 15 K results in an uncertainty of approximately 0.35 K.

[Figure]

The very good (average) agreement shown in Fig. 8 between 2 units frequently calibrated with LN2 is very encouraging, however having so many calibrations is not typical. Chances are that with one calibration every 6 months or less the K-band may be biased.

- That is why it is important to calibrate regularly in order to minimize these biases in the K-band. But we also showed that there is not too much drift anyway.
- After 15th June there was no additional calibration.

A second comment relates to the mention of tip curves. Although not the focus of the paper, in my opinion tip curves should be at least mentioned (if not discussed) as a mitigation strategy for long term drifts in between LN2 calibrations. With tip curves the "cold" target is an independent measurement of the sky and is therefore much lower than the LN2 temperature. Although their practical implementation is not straightforward, continuous tip curves, properly processed and monitored are probably the most accurate calibration for the K-band and can well capture drifts and fluctuations in the receiver hardware. Therefore, for completeness of discussion, especially for users non entirely familiar with the instrument, there should be at least a mention of calibration with tip curves as a mitigation strategy for long term drifts between LN2 calibrations.

- We added this and shortly discuss tip curves now in Section 5 at lines 683f as you suggested here.

**Minor comments:**

Section 4.2.1, Line 400:

"We know, however, that this particular calibration has to be faulty, as indicated by the large jumps in OmB within the K-band after this event (see Figure 5a and Section 4.2.2 for more details on OmB)." I may be misinterpreting here, but if I look at Fig. 5 where there is the dashed line on July 22 the OmB actually decreases after the calibration suggesting that it was somewhat successful. What am I missing?

- There shouldn't be OmB data points on the dashed line, as on the day of the calibration daily OmB means don't make sense (because on that day there will be measurements before and after the calibration, which should not be averaged). We deleted those data points in Fig. 5 on the days of calibration.
- You are, however, still correct in your assumption that the OmB immediately after the calibration on July 2022 seems to be looking ok, indicating a somewhat successful calibration. We removed the faulty calibration assumption here and rephrased the section 4.2.1 after line 399 (as well parts of section 4.2.2) accordingly. A short while after the calibration in Jul. 2022 we still see a large deviation in OmB though, indicating something is going wrong with the instrument.

Section 4.2.2, Line 476:

"Our analysis shows that discerning long-term drifts with OmB statistics is challenging." This is certainly true when using model output. If co-located radiosondes are available, uncertainty due to balloon drifts or spectroscopy are generally much smaller than calibration biases and drifts.

- This is correct. Co-located radiosondes are, however, mostly not available for the general operator. So using model output is the more feasible option for them in trying to discern long-term drifts.
  We added such a statement in Section 5 at lines 683f.